**Data Availability Statement:** Some of the data set, specifically the individual data points generated

# Development of an observational exposure human biomonitoring study to assess Canadian children's DEET exposure during protective use

**Jennifer C. Gibson**[1]* , **Leonora Marro**[1], **Michael M. Borghese**[1], **Danielle Brandow**[1], **Lauren Remedios**[1], **Mandy Fisher**[1], **Morie Malowany**[1], **Katarzyna Kieliszkiewicz**[2‡], **Anna O. Lukina**[1‡], **Kim Irwin**[3‡]

1 Environmental Health Science and Research Bureau, Health Canada, Ottawa, Ontario, Canada, 2 Pesticide Laboratory, Regulatory Operations and Enforcement Branch, Health Canada, Ottawa, Ontario, Canada, 3 Pest Management Regulatory Agency, Health Canada, Ottawa, Ontario Canada

☯ These authors contributed equally to this work.
‡ These authors also contributed equally to this work
* jennifer.gibson@hc-sc.gc.ca

## Abstract

Biomonitoring data of *N,N*-diethyl-*meta*-toluamide (DEET) in children is scarce and limited to controlled exposure and surveillance studies. We conducted a 24-hour observational exposure and human biomonitoring study designed to estimate use of and exposure to DEET-based insect repellents by Canadian children in an overnight summer camp setting. Here, we present our study design and methodology. In 2019, children between the ages of 7 and 13 took part in the study (n = 126). Children controlled their use of DEET-based insect repellents, and provided an account of their activities at camp that could impact insect repellent absorption. Children provided a total of 389 urine samples throughout the study day, and reported the time that they applied insect repellent, which allowed us to contextualize urinary DEET and metabolite concentrations with respect to the timing of insect repellent application. DEET (2.3% <Limits of detection (LOD)) and two metabolites, *N,N*-diethyl-*m*-(hydroxymethyl)benzamide (DHMB) (0% <LOD) and 3-diethylcarbamoyl benzoic acid (DCBA) (0% <LOD), were measured in urine samples. Three time difference scenarios were established for the data and analysed to account for these complex time-dependent data, which demonstrated the need for DEET biomonitoring to be done in context with the timing of a known DEET exposure or over the course of at least 14 to 24 hours to better capture the excretion curve. To our knowledge, this is the first field-based study of real-world exposure to DEET in children. Our experience and results suggest that this type of real-world observational exposure study with a human biomonitoring component can generate data reflective of actual exposure, but is not without significant logistic, practical, and analytical challenges.

from the children who participated in this study, are not available to be shared. These data are held in trust for the participants, who ultimately own their private data. Application for access to the data may be made to the Health Canada and Public Health Agency's Research Ethics Board via their email reb-cer@hc-sc.gc.ca. The project file number "REB 2018-019H" should be referenced in any inquiries. The consent process was clear in the restrictions placed around the data accessibility in order to protect the private information of the parents and children who participated in this study. These data underlie the means, standard deviations, and other measures reported about the participants in the study. It was assessed by Health Canada's Privacy Management Division that, due to the small sample sizes at certain sampling locations and within certain variables (i.e., age), it could only require the association of two or more variables for someone associated with that location but not participating in the study, to identify individuals from the data we collected. Thus the data carry a serious possibility of re-identification. The only minimal data set provided publicly will be the data underlying the quality assurance and quality control (QA/QC) testing for the viability of the study. Due to ethical restrictions and to protect participant privacy, non-identifying data may be made available to researchers only upon explicit request.

**Funding:** The authors received no specific funding for this work.

**Competing interests:** The authors have declared that no competing interests exist.

## Introduction

*N,N*-diethyl-*meta*-toluamide (DEET) is one of the most common and effective active ingredients in personal insect repellents. Health Canada recommends using insect repellents containing DEET to protect against mosquito and tick bites, which could lead to vector borne diseases including Lyme disease and West Nile virus [1]. In Canada, DEET-based insect repellents approved for sale are available with concentrations up to 30% DEET. Health Canada requirements state that children aged 2 to 12 years only use products containing 10% DEET or less, a maximum of three times daily [1]. Other jurisdictions, such as the United States, have different regulatory requirements for the use of DEET by the general population. Children of any age in the United States are permitted to use any concentration of DEET product (4–100%), and there are no restrictions on the concentration of DEET permitted to be sold [2,3].

Between 3.8% and 17% of DEET applied to skin is absorbed within hours of application and is rapidly excreted via urine [4–6]. DEET is metabolized in the body when absorbed, resulting in the creation of a primary metabolite *N,N*-diethyl-*m*-(hydroxymethyl)benzamide (DHMB) [7]. Further metabolism leads to the formation of a secondary metabolite 3-diethyl-carbamoyl benzoic acid (DCBA) [8–10].

DEET and its metabolites have been measured in children in several studies [9,11–13] including the United States' National Health and Nutrition Examination Survey (NHANES). In the 2011–2012 NHANES survey cycle, urinary DEET concentrations in children (age 6 to 19 years) were below the limit of detection for the whole population, although in 2013–2014, the 95th percentile had detectable concentrations [11]. Urinary concentrations of DCBA (creatinine adjusted) were detectable throughout the general population in the NHANES sampling years 2011 to 2016 [11], and were detected in 84% of the general population in a re-analysis of NHANES samples from 2007–2010, including children's samples [12]. In Australia, Heffernan et al. [13] analysed surplus, archived urine, collected as a part of routine medical testing, for DEET and its metabolites. While DEET was only detected in 17% of the sex- and age-stratified pooled samples, representing all ages (0->60 years) of the general Australian population (n = 2400), DCBA and DHMB were detected in all sex- and age-stratified pooled samples. Both of these instances of surveillance biomonitoring are representative of a background or incidental measurement of DEET and metabolites in a population, rather than a targeted characterization of levels present when DEET-based insect repellents are intentionally applied.

Tian and Yiin [9] examined the urinary metabolites of DEET in 17 children and 9 adults who were exposed to a single controlled application of 12% DEET to the skin of the arms and legs. Urine was collected from the participants over the course of the next 8 hours following application, and DEET, DCBA and another metabolite were detected in most of the urine samples. This was a controlled exposure study, which is useful for understanding the kinetics of DEET exposure, but does not provide insight into children's real-world use of insect repellents.

The objective of this study is to address this knowledge gap about children's real-world use of DEET-based insect repellents, by generating more modern, relevant Canadian exposure data. We conducted a 24-hour observational exposure and human biomonitoring study designed to estimate use of, exposure to, and excretion of DEET-based insect repellents by Canadian children in an overnight summer camp setting. This paper outlines the approach taken to generate this biomonitoring data, as well as the necessary statistical approaches applied to analyse the excretion of DEET and metabolites within the context of the timing of insect repellent application.

## Materials and methods

This project and all materials associated with the project have been found to meet ethical requirements for research involving humans by the Government of Canada's Health Canada and Public Health Agency of Canada Research Ethics Board (REB). The project has retained a Certificate of Ethics Review under Project File Number REB 2018-019H since November 2018.

Informed written consent was obtained from parents/guardians of participants, and informed written assent was obtained from children before the study commenced.

### Pilot study

A 7-hour long pilot study was conducted in March, 2019 with a convenience sample of 14 children (aged 7 to 12 years) to test the feasibility, acceptance and tolerance of study procedures during a simulated camp day. The purpose of the pilot study was to validate the study protocol, streamline our participant intake and data collection processes and identify and minimize the foreseeable challenges with collecting and storing urine samples while maintaining quality assurance in a real-world setting.

Children took part in supervised activities throughout the day, including an outdoor hike, which were designed to simulate overnight summer camp activities. Prior to the outdoor hike, the participants were offered an opportunity to apply insect repellent, if desired.

Participants provided full void urine samples on an as needed basis throughout the day. At the end of the pilot study day, the participants completed a feedback questionnaire on the clarity and ease of use of the study materials and urine sampling instructions.

### Main study

Volunteer participants (target ages 6 to 12 years) were recruited from among the incoming clients of three Ontario overnight summer camps. Camps aided in the outreach to parents, but the principal investigator and study coordinator were responsible for ensuring parental consent was received.

Facilities available for the study varied depending on the camp site. At one camp site, all urine collection was done in an indoor bathroom facility, and study staff had access to a dedicated refrigerator for sample storage. At the two other camp sites, urine was collected in either outhouses or indoor facilities, and the samples were cooled in a refrigerated cooler prior to storage in a cooler with ice packs. For transport to the Health Canada Pesticide Laboratory, all urine samples were stored in coolers with ice packs.

**Data collection.** In advance of the 24-hour study day, parents were provided with either an electronic or physical copy of the informed written consent document and the parental questionnaire. The questionnaire collected health and age information for the child participant, household socio-economic information, and family insect repellent habits (S1 File). The health information was scrutinized by study staff for considerations with respect to interactions with the children as well as potential contraindications for DEET exposure, based on anticipated known childhood issues (e.g., asthma, allergies), and impacts or contraindications documented in the literature (e.g., seizures/epilepsy, skin irritations/eczema/rosacea, kidney problems) [14–17]. Conversations were initiated with parents to ensure that parents were certain about enrolling the child into the study.

At the campsite, in the evening prior to the study start date, study staff obtained the informed written assent of participants. The participants' height (cm), and weight (kg), as well as the mass (g) of and information about the participant's insect repellent product were recorded at this time. Product information included the following: repellent brand, type of

applicator (e.g., lotion, pump, aerosol spray), registration number, percentage of DEET in repellent, and the last time they used the repellent prior to the study day (yesterday, 1 to 2 days ago, last week, never). Participants that did not have their own DEET-based insect repellent were provided with an insect repellent containing 7% DEET (OFF!® familycare®, SC Johnson) for the duration of the study. Participants were directed to use their insect repellent as they would normally. Study staff kept the participants' activity journals (S2 File) at the central study location setup, or closest to the bathroom facilities used during the study period for ease of access.

Participants provided urine samples between 7:00 and 22:00 on the study day without any restrictions, including on the timing or frequency of collection. All participants were offered the specimen jar for direct urine collection while females were also offered the option to urinate into a pre-placed, clean, single use, in-toilet urine specimen collection pan (Uri-Pan commode specimen collector, Plasti-Products Inc.). Once the participant exited the bathroom, a study staff member would enter and transfer the urine from the specimen collection pan into a clean urine sample cup (multi-purpose 500mL polypropylene container, Sarstedt). The lids of the urine sample cups were sealed with Parafilm™ M wrapping film (thermoplastic) to prevent spillage during transport. The sealed urine sample was labelled and secured inside a resealable poly bag. The study identification number (ID), sample number, time and date were noted on the chain of custody sheet, and the sample was placed in a cooler with ice to store at 4°C. A study by Smallwood et al. suggested that concentrations of the DEET parent compound were found to remain constant for 8 days in urine when stored at 4°C [18].

At the time of each urine sample collection, study staff assisted the participant with completing their activity journal, which tracked daily behaviour and insect repellent application. Participants indicated if, when, and where on their body insect repellent was applied; whether hand sanitizer or sunscreen had been used in concert with insect repellent application; and hand washing habits post-insect repellent application.

Quality assurance and quality control (QA/QC) samples were created during the study day. Pre-mixed aliquots of low, medium, and high concentration DEET (1.0, 8.0, 128 ng/mL) (N,N-(Diethyl-d10)-m-toluamide, Toronto Research Chemicals), DHMB (0.2, 1.6, 25.6 ng/mL) (N,N-Diethyl-m-hydroxymethylbenzamide-D10, Toronto Research Chemicals), and DCBA (2.0, 16, 256 ng/mL) (3-[(Diethylamino)carbonyl]benzoic acid, Toronto Research Chemicals) in blank urine samples derived from an adult male volunteer who was never exposed to DEET, were kept cool from the laboratory to the field. An aliquot of each concentration level for each compound was transferred from a 50mL polyethylene Falcon tube (Corning™ Falcon 50mL Conical Centrifuge tubes) to a clean urine sample cup, and the lid was secured with Parafilm™. A distilled water blank was created in a similar manner. This occurred in the morning, afternoon, and evening of the study day, resulting in a total of 12 QA/QC samples being created per study day. The QA/QC samples were stored with the urine samples to ensure consistent handling of all sample containers.

On the morning after the study day, beginning at 7:00, a morning void was requested and the final insect repellent container mass (g) was recorded. Participants completed the final questions in their activity journal to describe: if the participant urinated during the study period but did not collect the urine sample; if the participant showered, sweated from exercise, or swam in the morning, afternoon or evening of the study day; and lastly, if they spilled or shared their insect repellent, or used anyone else's insect repellent.

All data collected (i.e., anthropometric measurements, data from activity journals, and sample times) were de-identified from a participant's name and associated solely with a participant study ID. The data were input separately by study staff into two identical copies of a purpose-built database, which was cross-checked for discrepancies.

**Analytical methods for urine samples.** Chemical analysis of urine samples was performed by the Health Canada Pesticide Laboratory for DEET, DCBA, and DHMB. Samples were incubated at 37°C overnight in the presence of β-glucuronidase (*Helix pomatia*) to deconjugate the glucuronide-bound analytes. The samples were then carried through a solid-phase extraction using a Waters Oasis HLB 96-well plate and reconstituted in aqueous methanol prior to analysis by LC-MS/MS (Shimadzu UPLC-AB Sciex QTRAP 6500+) [19]. Urine specific gravity was measured using an Atago PAL-10S handheld refractometer. Performance of the analytical methods was tested through the use of internal reference samples and field QA/QC spike and blank samples. The limits of detection (LOD) were: DEET (0.27 μg/L), DHMB (0.038 μg/L), and DCBA (0.41 μg/L). Any samples returning results below the limit of detection were replaced in the data set by LOD/2.

Creatinine analysis was done at a Health Canada Health Products and Foods Branch laboratory. Creatinine was measured in urine using the colorimetric end-point Jaffe kinetic method. An alkaline solution of sodium picrate reacts with creatinine in urine to form a red Janovsky complex. The absorbance was read at 510nm on a Horiba Medical ABX Pentra 400 chemistry autoanalyser [20].

Both specific gravity and creatinine measurements were taken to allow the samples to be standardized by either measurement of urine dilution. The data were standardized using both measurements of urine dilution, creating two data sets, and statistical analyses were completed on both data sets.

**Analytical decisions, definitions, and assumptions.** Since participants were free to apply their insect repellent and participate in urine collection at any time throughout the study (i.e., as they would normally at overnight camp), it was not possible to standardize the timing of the insect repellent application and urine collection times. Our analysis of these complex time-dependent data and our description of the distribution of DEET, DHMB, and DCBA are driven by the following definitions, and assumptions.

We used the time difference between the urine sample collection and the most recent application of insect repellent to describe the exposure distribution of DEET, DHMB, and DCBA. This distribution depended on two relatively uncertain (in children) toxicokinetic factors: 1) the amount of time needed for DEET exposure resulting from an insect repellent application to be metabolized by the body and excreted in a urine sample—referred to here as the lag of time (in hours); and 2) the length of time that DEET and metabolites may be excreted and detected following exposure—referred to here as the width of the time intervals in the distribution.

We used three different lag times to represent the time between DEET application and subsequent detection in urine. Our initial consideration was based on an "unrestricted" assumption that DEET levels from the most recent application time prior to a urine sample (regardless of whether the last application occurred as long as 12 hours or immediately prior) could be detected in the urine sample and this was labelled a lag of 0 hours (lag0). Although this is somewhat unrealistic, we felt that it was a reasonable initial assumption to begin with no time lag adjustment. We then considered two lag times of 2 hours (lag2) and 4 hours (lag4), which were chosen based on previous studies in human volunteers that detected DEET in blood and urine within these time frames [6]. Under lag2, insect repellent applications that occurred 2 hours or more prior to the urine sample collection time were assumed to be detectable in urinary DEET concentrations. If an application happened 2 hours or less before a urine sample time, we ignored that insect repellent application for the purposes of that urine sample, and considered the previous application time to have impacted the urine sample. If there was no previous insect repellent application within the 24-hour study period, then that urine sample would become part of a "baseline" group representing minimal or background exposure. We applied the same decisions using 4 hours (lag4).

**Table 1. Post-application time intervals chosen for urine sample data analysis.** Intervals were built to organize urine samples into similar groupings of time since insect repellent application.

| Lag[1] (hours) | Interval width[2] (hours) | Time intervals post-application | | | |
|---|---|---|---|---|---|
| | | 1 | 2 | 3 | 4 |
| **0** | 6 | 0–≤6 | >6–≤12 | >12–≤18 | >18–≤28 |
| **2** | 6 | 2–≤8 | >8–≤14 | >14–≤20 | >20–≤28 |
| **4** | 6 | 4–≤10 | >10–≤16 | >16–≤22 | >22–≤28 |

[1] Amount of time needed for DEET exposure resulting from application to be detected in a urine sample.

[2] Length of time that DEET and metabolites may be detected in urine samples following exposure.

The width of the time intervals in the distribution was based on the estimated half-life of DEET in the human body, 8 to 12 hours [6,21]. To capture exposure respecting this estimated half-life time we chose an interval width of 6 hours. These 6-hour intervals were incorporated with the lags to provide a range of time-windows of detection across the 24-hour study period, resulting in 4 time windows for each lag (Table 1). Following this approach, we were able to examine the distribution of DEET and its metabolites using multiple time-windows of exposure and metabolism. The final interval was extended for lag0 and lag2 to facilitate comparison with lag4 because few urine samples were collected after 24-hours.

The baseline group consisted of urine samples where either an insect repellent application was not used prior to the urine sample (lag0) or the only insect repellent application prior to urine sampling was within a lag time period during which it would not be expected to influence the urine sample in lag2 and lag4, *i.e.*, a repellent application at 9:30 and a urine sample at 10:00 (within 2 or 4 hours of the urine sample). Baseline sub-groups were created for urine samples where the application of repellent was categorized under "yesterday", "1 to 2 days ago", "last week" or "never".

Fig 1 and Table 2 exemplify these concepts using data from a random participant in our study population. One urine sample occurred prior to the initial application of insect repellent. Since the most recent repellent application for this sample was "yesterday", it was assigned to the baseline group "yesterday". At 09:30, they applied insect repellent followed by a urine sample collected at 13:50. Another application of insect repellent occurred at 17:30 and another urine sample collected at 18:00. The final insect repellent application was at 20:00 with urine samples collected at 21:00 and the following morning at 06:30. Assuming a lag of 0 (*i.e.*, the application immediately prior to the urine sample is assumed to be detected in the urine sample), then the time difference for each urine sample would be as follows: 4 hours 20 minutes, 30 minutes, 1 hour, and 10 hours 30 minutes (Table 2). The corresponding time interval for the distribution would place the second, third and fourth urine samples in the 0 to 6 hour time interval, and the final urine sample in the 6 to 12 hour time interval. Assuming a lag of 2 hours (*i.e.*, the application that occurred 2 or more hours prior to the urine sample is assumed to be detected in the urine sample) then the time difference for the same urine samples would be: 4 hours 20 minutes, 8 hours 30 minutes, 3 hours 30 minutes, and 10 hours 30 minutes. In this case, the insect repellent application that occurred prior to the third and fourth urine samples was within the 2 hour lag time, and therefore the previous insect repellent application was used to determine the time since application. The corresponding time interval for the distribution would place the second and fourth urine samples in the 2 to 8 hour time interval and the third and final urine samples in the 8 to 14 hour time interval.

**Statistical analysis.** DEET, DHMB and DCBA concentrations were standardized for specific gravity (SG) and creatinine (CR). To standardize concentrations for SG, we multiplied

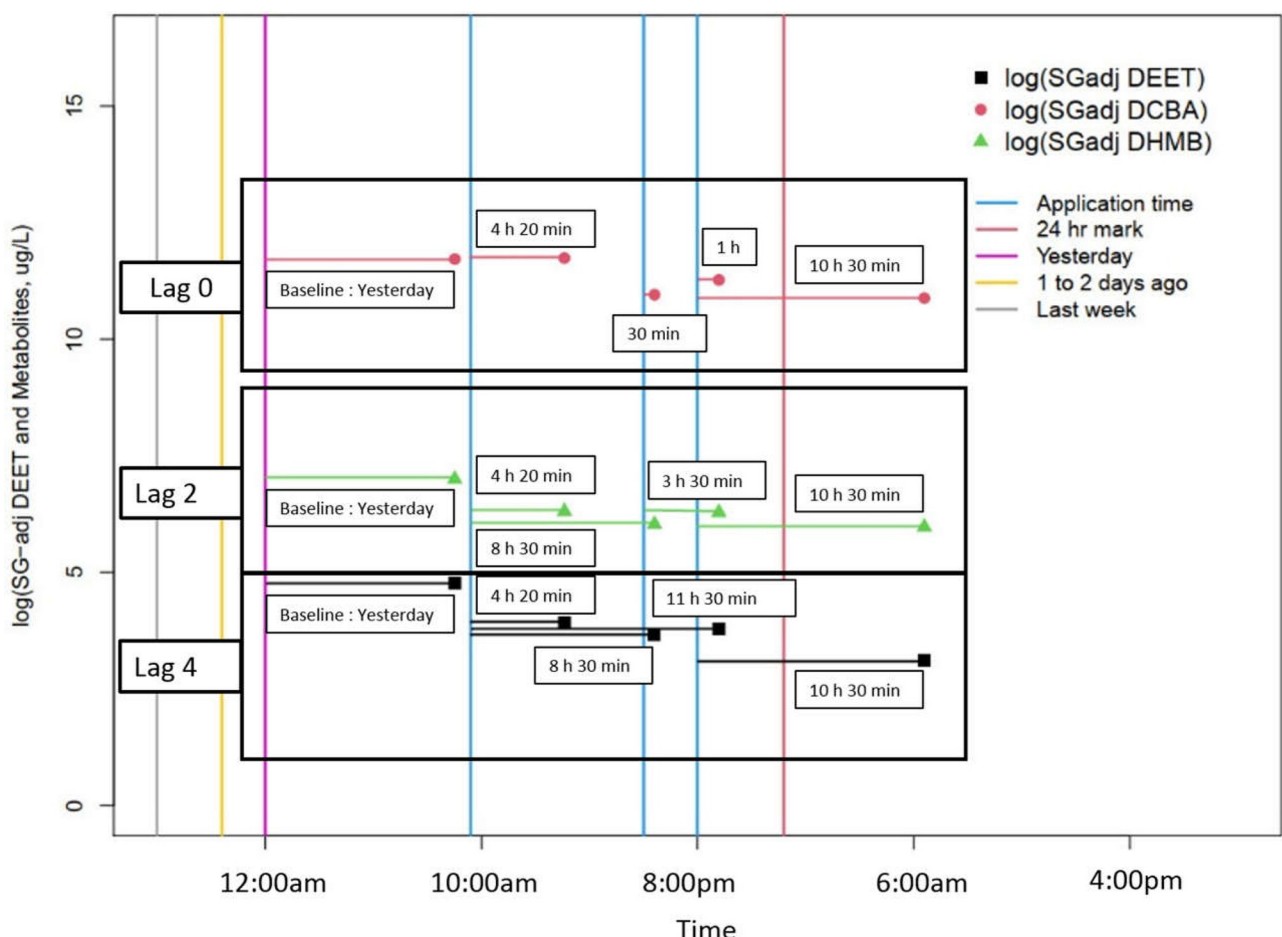

**Fig 1. Profile of a participant during the 24-hour study period, including timing of insect repellent applications and urine samples.** Specific gravity standardized concentrations of DEET, DHMB, and DCBA are presented as points.

**Table 2. Organization of urine samples for a single participant according to the three time difference scenarios constructed.**

| Urine Sample | Time of insect repellent application | Time of urine sample | Lag 0 | | Lag2 | | Lag4 | |
|---|---|---|---|---|---|---|---|---|
| | | | Time since application | Time interval for distribution | Time since application | Time interval for distribution | Time since application | Time interval for distribution |
| | Yesterday | | | | | | | |
| 1 | | 8:45 | Baseline Yesterday | Baseline Yesterday | Baseline Yesterday | Baseline Yesterday | Baseline Yesterday | Baseline Yesterday |
| | 9:30 | | | | | | | |
| 2 | | 13:50 | 4hrs20min | 0 to 6 | 4hrs20min | 2 to 8 | 4hrs20min | 4 to 10 |
| | 17:30 | | | | | | | |
| 3 | | 18:00 | 30min | 0 to 6 | 8hrs30min | 8 to 14 | 8hrs30min | 4 to 10 |
| | 20:00 | | | | | | | |
| 4 | | 21:00 | 1hr | 0 to 6 | 3hrs30min | 2 to 8 | 11hrs30min | 10 to 16 |
| Next day | | | | | | | | |
| 5 | | 6:30 | 10hrs30min | 6 to 12 | 10hrs30min | 8 to 14 | 10hrs30min | 10 to 16 |

the observed metabolite concentration urine sample ($P_i$) by the following formula adapted from Duty et al. [22].

$$P_c = P_i \left[ \frac{SG_m - 1}{SG_i - 1} \right]$$

Where $P_c$ is the SG-standardized metabolite concentration, $SG_i$ is the specific gravity of the observed urine sample, and $SG_m$ is the median SG for the cohort. To standardize concentration for CR, we divided the observed metabolite concentration urine sample ($P_i$) by its corresponding creatinine observation. Urine levels of DEET, DHMB and DCBA are log-normally distributed; therefore, geometric means and corresponding confidence intervals were reported for the raw, SG-standardized, and CR-standardized metabolite values. Linear mixed effect models (LMM) were used to characterize the data and account for multiple urine samples per participant, where participants were treated as random effects. LMM, adjusted for time intervals, were used for "in study" plus baseline groups in order to compare all groups to the baseline group "Yesterday", as well as comparison of "in study" time interval groups to the final time interval ending at 28 hours. One measurement was missing for a participant's weight, which was imputed based on the participant's height and the weights of all participants of the same sex at birth. Pairwise tests were carried out to compare all time intervals to the baseline group "yesterday" and then again to compare the "in study" samples to the final time interval ending at 28 hours. Bonferroni adjustments were used to account for multiple pairwise comparisons. Analyses were conducted using SAS Enterprise Guide v. 7.1 and statistical significance was specified as a p-value of less than 0.05.

## Results

### Pilot study

Of the 14 children enrolled in the pilot study, one participant did not provide urine samples or complete their activity journal and was, therefore, not included in the pilot analysis. The 13 remaining participants completed the activity journal and participant questionnaire, although only 8 of those participants provided a total of 10 urine samples. Most of the participants (92%) indicated that the activity journal was easy to complete and that the instructions for collecting urine samples were adequate. Procedures were generated to account for urine sampling away from the central location, such as packing urine cups and bags so that they could be taken by a participant on a hike.

The pilot study QA/QC field spike samples recovery ranged from 76 to 103% of the expected concentrations. Urine concentrations were measured in the lab, but the data were not incorporated into the study dataset because the pilot study (7 hours) was too short compared to the main study (24 hours).

### Main study

There were 154 participants initially registered in the study. However, 28 participants withdrew from the study. For the 126 participants retained, the median age was 11 years (range 7 to 13 years), and 124 provided at least one urine sample.

A total of 391 urine samples were collected and analysed from 124 participants, with an equal number of samples from female and male participants (Table 3). Two urine samples were suspect and were discarded from the final analysis leaving a total of 389 urine samples. Nine samples had DEET concentrations below the LOD, which were collected from 5 individuals in two of the three summer camps. Both metabolites were detected in 100% of samples.

**Table 3. Descriptive characteristics of urine samples (n = 389) and the distribution of urine samples into lag times.**

| Variable | Group | N | % |
|---|---|---|---|
| Sex at birth | Female | 192 | 49.4 |
| | Male | 197 | 50.6 |
| Morning void sample | No | 311 | 80.0 |
| | Yes | 78 | 20.0 |
| Number of samples provided | 0 | 2 | 1.6 |
| | 1 | 21 | 16.7 |
| | 2 | 24 | 19.0 |
| | 3 | 26 | 20.6 |
| | 4 | 22 | 17.5 |
| | 5 | 30 | 23.8 |
| | 6 | 1 | 0.8 |
| **Distribution of urine samples** | | | |
| Lag0 | Baseline | 165 | 42.4 |
| | In Study | 224 | 57.6 |
| Lag2 | Baseline | 215 | 55.3 |
| | In Study | 174 | 44.7 |
| Lag4 | Baseline | 237 | 60.9 |
| | In Study | 152 | 39.1 |

The majority of the samples collected were taken during the study day period, with only 20% of samples collected at the morning void the day after the study. Most participants (83%) gave more than one urine sample during the course of the study.

Concentrations of DEET were the lowest of the three compounds measured, but detectable in the majority of participants (Fig 2). DHMB concentrations were 3- to 6-times higher than DEET concentrations. DCBA concentrations were 28- to 60-times higher than DHMB concentrations. As seen in Fig 2, DEET concentrations are highest in the initial time intervals for each lag, encompassing >6–≤12 hours post-application at lag0, >2–≤8 hours post-application at lag2 and >4–≤10 hours post-application at lag4, and decline monotonically with time thereafter. Metabolites DHMB and DCBA are elevated in the 0–≤6 hour time interval in comparison with the Baseline intervals, but seem to reach peak concentrations in the later time intervals. The distribution of DHMB with a lag0 shows a gradual increase to a peak concentration at the >18–≤28 hour interval, whereas the distribution of DCBA with a lag0 shows a peak concentration at the >6–≤12 hour interval. However, the distribution of both DHMB and DCBA with a lag2 shows a peak concentration at the >8–≤14 hour interval, which is similar to the distribution of both DHMB and DCBA with a lag4, where the peak concentration occurs at the >10–≤16 hour interval. In both of these latter time difference scenarios, DHMB and DCBA decrease after this peak concentration. Histograms depicting the distribution of the geometric means of the creatinine standardized concentrations of DEET, DHMB and DCBA are in S1 Fig.

**Main study QA/QC.** Three sets of field QA/QC samples, which included a distilled water blank, and low, medium and high concentration spikes of combined DEET, DHMB, and DCBA, were created for each study day in the morning, afternoon, and evening sampling times (n = 92). The QA/QC samples were stored and transported with the urine samples. Ideal percent recoveries fall within 80 to 120% of the expected concentration, as the sample urine concentrations measured were 3 times higher than the LOD [19]. DEET is the only compound

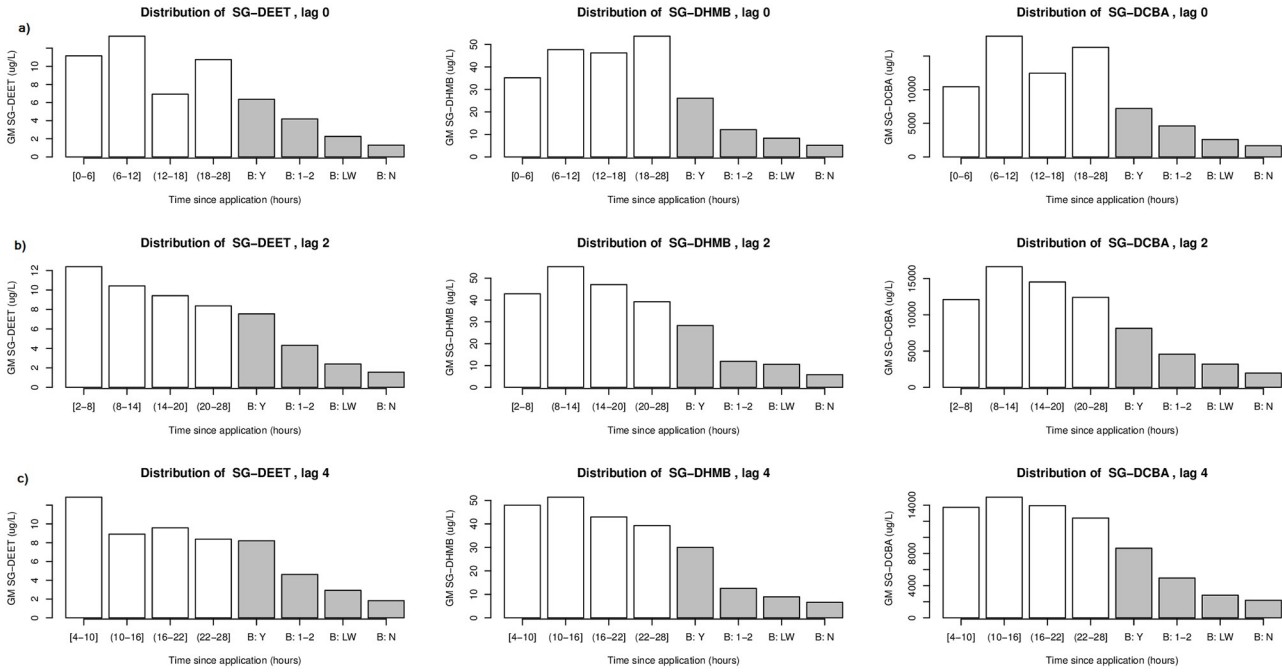

**Fig 2. Distribution of geometric means of DEET, DHMB, and DCBA (specific gravity (SG) standardized) according to the three data manipulation scenarios constructed—lag0, lag2, and lag4.** (a) distribution with a lag of 0 hours, (b) distribution with a lag of 2 hours, (c) distribution with a lag of 4 hours. (B:Y—Baseline group: Yesterday; B:1–2—Baseline group: 1–2 days ago; B:LW—Baseline group: Last week; B:N—Baseline group: Never.).

we would expect to potentially contaminate samples in the field, as DEET-based insect repellent was being used around the site. Aside from bodily metabolism, there are no sources of DHMB or DCBA at the study sites.

There were 12 instances (52%) of DEET levels in excess of 120% of expected values being recovered in the low concentration DEET field spikes (expected: 1 μg/L; exceedances range: 1.52–4.43 μg/L). There were 6 instances (26%) of DEET levels in excess of 120% of expected values being recovered in the medium concentration field spikes (expected: 8 μg/L; exceedances range: 9.62–23.82 μg/L) (S1 Table). And there were 10 instances (43%) of low DEET levels being detected in field blanks (range: 0.32–1.14 μg/L) (S2 Table). The levels found in these field spikes may indicate some incidental transfer from the environment. There were also 11 instances (16%) of low and medium concentration DEET field spikes returning recoveries below 80% of expected values. The majority (93%) of the high concentration DEET field spikes fell within the acceptable recovery margins, with 5 field spikes returning recoveries below 80% of expected values (S1 Table). Urine sample data were inspected and one anomalously high (9725 μg/L) DEET concentration resulted in that urine sample being dropped from the data set, as it was likely due to direct contamination by insect repellent.

With the exception of a single sampling event at the campsite with a dedicated refrigerator for the study, the majority (85.5%) of the low, medium, and high concentration DHMB field spikes showed low percentage recoveries (S1 and S3 Tables). This could indicate degradation of DHMB in the field. An examination of the differences between urine samples grouped by campsite indicated no significant differences between camps in DEET or DHMB concentrations in urine (S4 Table).

DCBA had the most consistent and acceptable recovery percentages. There were 10 instances (43%) of DCBA returns exceeding 120% of expected values in the low concentration

QA/QC samples (expected: 2 µg/L; exceedances range: 2.87–10.79 µg/L). There were 7 instances (30%) of returns exceeding 120% of expected values in the medium concentration QA/QC samples (expected: 16 µg/L; exceedances range: 19.4–22.92 µg/L) (S1 Table). And there was one instance of DCBA detection in a field blank (7.8 µg/L), the source of which was undetermined (S2 Table). It is unlikely that there would be a source of DCBA at the campsite, as it is a metabolite of DEET, and it is unlikely that it was due to transfer from a field spike spill, as there was no DEET or DHMB detected in that blank. All high concentration QA/QC field spikes for DCBA had recoveries within acceptable margins. The examination of the differences between urine samples grouped by campsite did not indicate an obvious pattern in DCBA urine concentrations related to camp, unless the day of camp or dose exposure was taken into account. In this case, the camp that was becoming significantly different was one of the two camps reliant on coolers for sample preservation.

## Discussion

This is the first known human biomonitoring study examining the real world use of DEET based insect repellents in children. Using materials designed for child engagement and readability (S2 File), we collected urine samples as well as data on insect repellent use and daily activities, from a population of 124 Canadian children 7 to 13 years of age. The urine sample analyses generated data for excreted DEET with a 97.7% detection rate, and 2 DEET metabolites, both with 100% detection rates.

Both specific gravity and creatinine were used to adjust the urine samples for dilution in our analyses. Creatinine levels are influenced by child age, body mass index, and season, whereas specific gravity is much more stable [23,24]. Therefore, specific gravity is considered a more robust method for adjusting for urinary dilution in children. One study of 243 children aged 8 to 11 years found that specific gravity showed a higher temporal consistency across 5 consecutive weekdays compared to creatinine [24]. However, many studies that would be a source of comparison opt to adjust for urine dilution by creatinine, for example, NHANES and the Canadian Health Measures Survey. Therefore, to preserve the ability to make these comparisons in the analysis, we also created a set of creatinine standardized urine data.

We used three time difference scenarios for the data to compare urine samples in the equivalent stage of metabolism and excretion of a given insect repellent application. The shortest time interval examined in previous studies was 2 to 4 hours ([6]—human plasma 2 hours, urine 4 hours), ([25]—human urine, selected 4 hour compartments), ([26]—human urine 4 hours), which led to the choice of investigating a lag time of 0, 2 and 4 hours. The processes for the excretion of DEET metabolites in urine appear to require at least a short time lag between application and excretion [6,25,26]. The analysis of a zero lag time was included to take into account the possibility of immediate excretion. Plasma half-lives of DEET range from 2.5 to 9 hours in animal studies [27–30]. In the controlled dosing study, most of the absorbed radioactivity associated with the labelled DEET applied to adult study volunteers was excreted within 12 hours of the application to skin [6]. In that same study, the peak plasma radioactivity associated with 15% DEET exposure was at 4 hours, with only a minor decrease by 6 hours [6]. As such, time intervals of six hours were chosen for the time difference scenarios, to visualize the half-life of DEET metabolism in participants.

Distinct peaks are visible in the histograms of the geometric means calculated for DEET, DHMB and DCBA urine concentrations in the first or second time intervals in the lag2 and lag4 scenarios, with gradually decreasing geometric mean concentrations of the compounds excreted over the remainder intervals (Fig 2). The lag0 scenario histograms contain no clear pattern during the "in study" timeframe for any of the compounds. For future investigations

into DEET and its metabolites, in order to capture the full expression of the metabolism and excretion of the compounds, a time window at least as long as 14 to 24 hours should be considered. Single spot urine samples, depending on the time context related to insect repellent use, may not provide the most accurate measure of DEET exposure, given the change over time in urine concentrations. Multiple urine samples over that 14 to 24 hour period would provide a much better measurement of DEET exposure.

Due to the uncontrolled setting, the QA/QC samples created in the field were imperative to monitor for contamination or degradation of samples. The low percentage recoveries of DHMB from the field spikes may indicate that there was some degradation of the DHMB metabolite during storage and transport. If so, the concentrations measured in the urine may be an underestimate of the actual concentration. There were no significant differences in DEET or DHMB concentrations in samples between the three camp sites, so any degradation that occurred was not sufficient to significantly skew the DHMB results from the DEET results. Smallwood et al. [18] focused solely on the stability of the parent compound DEET in urine samples and found that DEET remains stable in urine at 20˚C for at least 8 days. Internal stability studies (unpublished) indicated that DHMB should remain stable at room temperature (23–29˚C) for up to two weeks. Conditions in the field included a combination of sunny and overcast days, with recorded temperatures ranging from 20˚C to 30˚C, with an average temperature of 24.6˚C. But the DHMB recoveries averaged 58% overall, therefore, due to the poor recoveries of the DHMB QA/QC samples, the DHMB data will not be relied upon for future analysis.

Although the higher than expected DEET recoveries found in some of the field spikes and blanks indicated potential low level contamination, inspection of the urine results only required the removal of one urine sample from the data set, due to an anomalously high DEET concentration (9725 µg/L). In general, the DEET concentrations measured in urine were in line with or slightly below the medium concentration field spike, with the upper end of the observed urine concentrations exceeding the high concentration field spike. It was not considered necessary to adjust the urine concentrations to take into account any low level DEET contamination.

As the DCBA concentrations in urine were much higher than the concentrations in the field spikes, the main concern for the study would be detecting degradation of the DCBA field spikes during storage and transport, which was not detected. The low and medium field spike QA/QC samples with DCBA concentrations exceeding a 120% spike return were found at all three sampling locations, however the high concentration field spikes for DCBA all had recoveries between 80–120% of expected values. The consistent recoveries of the higher concentration spikes for DCBA provides confidence in the DCBA urine results.

While conducting an observational exposure and human biomonitoring study in a more uncontrolled setting, such as a real, operating overnight summer camp, introduced some complexities to the data, it was a valuable study design to gain insight into much more realistic exposure patterns. Participants were not directed on when to use insect repellent by study staff intentionally, and we believe this is a strength of the study design, because it reflects real-world insect repellent exposure patterns in children. Insect repellent application behaviour could have been influenced by the knowledge that the participants were taking part in a study investigating insect repellent use; however, the day-to-day activities at the camp and the environmental setting (i.e., insect pressure) were likely to be the greater determinants of insect repellent use as demonstrated by the clustering of insect repellent application times to early morning and evening hours. Several of the variables collected and described herein were used in further analyses outside of the scope of this paper, for example, the height and weight of the children were used to derive the body surface area to aid in estimates of dose exposure [31].

Several participants in the pilot study reported that they were uncomfortable with having their body weight data written in their journal where their peers would be able to see it. In response to this, we ensured only study staff would have a master list of participant identification numbers with the height and weight recordings, and we modified the activity journal accordingly. Additionally, the other participants were kept away from the measurement area when a participant was undergoing anthropomorphic measurements. After this change, we only had one participant refuse to have their weight measured, and that data point was imputed based on the participant's height and the weight of other participants of that sex.

We recruited 77% of our target population size, and retained 82% of participants. The restricted age range targeted by the study (6 to 12 years of age) created challenges in including more of the overnight summer camp client population. Certain locations only had restricted time periods when children of this age would be at camp. Also, it was noted by study staff that the older children were more likely to be engaged by the study and enticed to participate in the scientific inquiry. By necessity, the age range did increase to 7 to 13 year olds as some of the children recruited had just turned 13 prior to the study beginning. We were also not able to recruit 6 year olds for the study, either due to the limited age ranges available at camps, or a lack of interest on behalf of the parent or child.

## Conclusion

There is a great need to generate data that will support a better understanding of how children interact with their environment, medicine, and food [32,33]. When placed in a risk assessment context, the ability to consider a child-specific exposure can be hampered by this lack of data. The lack of child data can lead to the use of adult data adjusted by a calculated uncertainty factor to account for the physiological and behavioural differences of children [21,34]. To address this lack of child-specific data for DEET, this study was conducted.

It is possible to conduct an observational exposure and human biomonitoring study in children in an operational setting, such as an overnight summer camp. This study was successful in collecting 389 urine samples from 124 participants throughout a 24-hour study period, while also collecting behavioural information on DEET-based insect repellent use and other variables that may impact DEET absorption. Based on our investigation of different time lag scenarios, we conclude that a lag of at least 2 hours between insect repellent application and urine sampling is the most biologically realistic data aggregation.

Gathering human biomonitoring data in a real-world, free-living setting is complex but provides a rich distribution of samples and concentrations over time, and provides interesting insight into the exposure to, and metabolism of, DEET-based insect repellent by Canadian children.

## Supporting information

**S1 Fig. Distribution of geometric means of DEET, DHMB, and DCBA (creatinine (CR) standardized) according to the three data manipulation scenarios constructed—lag0, lag2, and lag4.** (a) distribution with a lag of 0 hours, (b) distribution with a lag of 2 hours, (c) distribution with a lag of 4 hours. (B:Y—Baseline group: Yesterday; B:1–2—Baseline group: 1–2 days ago; B:LW—Baseline group: Last week; B:N—Baseline group: Never).
(TIF)

**S1 Table. Number of QA/QC samples falling within the acceptable 80 to 120% recovery range, including number of samples falling outside that range.** a) Low concentration field

spikes. b) Medium concentration field spikes. c) High concentration field spikes.
(DOCX)

**S2 Table. Concentrations of detected compounds in QA/QC sample field blanks (µg/L).**
Limits of detection (LOD) are DEET (0.27 µg/L), DHMB (0.038 µg/L), and DCBA (0.41 µg/L).
(DOCX)

**S3 Table. Minimum, average, and maximum percent recoveries from all quality assurance and quality control (QA/QC) field samples.** Acceptable percent recovery ranges from 80 to 120% of expected concentrations, as sample concentrations exceed 3 times the limits of detection.
(DOCX)

**S4 Table. QA/QC samples by camp location.** Camp 1 had a dedicated refrigerator for samples for the duration of the study. Camp 2 and 3 relied on a refrigerated cooler to bring urine samples down to a cool temperature, and then stored urine with ice packs in coolers. a) Low concentration field spikes, anticipated concentrations: DEET 1µg/L; DHMB 0.2µg/L; DCBA 2µg/L. b) Medium concentration field spikes, anticipated concentrations: DEET 8µg/L; DHMB 1.6µg/L; DCBA 16µg/L. c) High concentration field spikes, anticipated concentrations: DEET 128µg/L; DHMB 25.6µg/L; DCBA 256µg/L.
(DOCX)

**S1 File. Health Canada DEET Study parental questionnaire, designed to gather child health data, socio-economic information for the household, as well as family habits with regards to insect repellent acquisition and use.**
(PDF)

**S2 File. Selected worksheets from the Health Canada DEET Study participant activity journal, designed to gather insect repellent information, application times, other personal care products used in concert with insect repellent, and body locations of insect repellent application.**
(PDF)

**S3 File. Abbreviations.**
(DOCX)

**S1 Data.**
(XLSX)

## Acknowledgments

The authors would like to acknowledge the participants and their families for their interest and participation in the study. The authors would also like to acknowledge the directors and staff of the three overnight summer camps for their enthusiasm and support for the study. Thanks to JAM and VH for their constructive feedback. The authors would also like to acknowledge the Health Library at Health Canada and the Public Health Agency of Canada for their high quality work in supporting this project.

## Author Contributions

**Conceptualization:** Jennifer C. Gibson, Leonora Marro, Michael M. Borghese, Lauren Remedios, Mandy Fisher, Morie Malowany, Anna O. Lukina, Kim Irwin.

**Data curation:** Jennifer C. Gibson, Leonora Marro, Danielle Brandow, Lauren Remedios, Morie Malowany.

**Formal analysis:** Jennifer C. Gibson, Leonora Marro, Michael M. Borghese, Danielle Brandow, Lauren Remedios, Mandy Fisher, Morie Malowany, Katarzyna Kieliszkiewicz.

**Investigation:** Jennifer C. Gibson, Michael M. Borghese, Danielle Brandow, Lauren Remedios, Mandy Fisher, Katarzyna Kieliszkiewicz.

**Methodology:** Jennifer C. Gibson, Leonora Marro, Michael M. Borghese, Danielle Brandow, Lauren Remedios, Mandy Fisher, Morie Malowany, Katarzyna Kieliszkiewicz, Anna O. Lukina, Kim Irwin.

**Project administration:** Jennifer C. Gibson, Danielle Brandow, Anna O. Lukina.

**Validation:** Jennifer C. Gibson, Danielle Brandow, Morie Malowany, Katarzyna Kieliszkiewicz, Kim Irwin.

**Visualization:** Jennifer C. Gibson, Leonora Marro, Danielle Brandow, Lauren Remedios, Mandy Fisher, Morie Malowany.

**Writing – original draft:** Jennifer C. Gibson, Leonora Marro, Michael M. Borghese, Danielle Brandow, Lauren Remedios, Mandy Fisher, Morie Malowany.

**Writing – review & editing:** Jennifer C. Gibson, Leonora Marro, Michael M. Borghese, Danielle Brandow, Lauren Remedios, Mandy Fisher, Anna O. Lukina, Kim Irwin.

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
