## [Decision Letter · Decision Letter 0]

3 Dec 2021

PONE-D-21-34278Development of an observational exposure human biomonitoring study to assess Canadian children's DEET exposure during protective use.PLOS ONE

Dear Dr. Gibson,

Thank you for submitting your manuscript to PLOS ONE. After careful consideration, we feel that it has merit but does not fully meet PLOS ONE’s publication criteria as it currently stands. Therefore, we invite you to submit a revised version of the manuscript that addresses the points raised during the review process.

ACADEMIC EDITOR: This is an interesting manuscript, and  based on the publication criteria of PLOS ONE's  several questions highlighted by the reviewers should be clarified by authors.

best regards .

Please submit your revised manuscript by  DEC, 30. If you will need more time than this to complete your revisions, please reply to this message or contact the journal office at plosone@plos.org. Please include the following items when submitting your revised manuscript:A rebuttal letter that responds to each point raised by the academic editor and reviewer(s). You should upload this letter as a separate file labeled 'Response to Reviewers'.A marked-up copy of your manuscript that highlights changes made to the original version. You should upload this as a separate file labeled 'Revised Manuscript with Track Changes'.An unmarked version of your revised paper without tracked changes. You should upload this as a separate file labeled 'Manuscript'.

We look forward to receiving your revised manuscript.

Kind regards,

José Luiz Fernandes Vieira

Academic Editor

PLOS ONE

Journal Requirements:

Reviewers' comments:

Reviewer's Responses to Questions

**Comments to the Author**

1. Is the manuscript technically sound, and do the data support the conclusions?

Reviewer #1: Partly

Reviewer #2: Yes

Reviewer #3: Yes

Reviewer #4: Partly

2. Has the statistical analysis been performed appropriately and rigorously? 

Reviewer #1: N/A

Reviewer #2: Yes

Reviewer #3: Yes

Reviewer #4: No

3. Have the authors made all data underlying the findings in their manuscript fully available?

Reviewer #1: Yes

Reviewer #2: Yes

Reviewer #3: Yes

Reviewer #4: No

4. Is the manuscript presented in an intelligible fashion and written in standard English?

Reviewer #1: Yes

Reviewer #2: No

Reviewer #3: Yes

Reviewer #4: Yes

5. Review Comments to the Author

Reviewer #1: The authors have done a lot of work for this field study, and earned some good results. The major issue, however, is the objective of this study, which is not very clear. The authors established three exposure scenarios for data analysis, but did not conduct a good comparison between them and suggest better ones (at least not seen in Abstract, but in Conclusion). The authors need to indicate the chosen scenarios explicitly by showing some good approaches of comparison, and make this paper more valuable.

Other issues are listed as follows:

Line 37, please give full names of DHMB and DCBA here.

DHMB is an intermediate product on the way to DCBA, and thus, inconsistent recovery rates for it are acceptable. Other than DCBA, a final product of ring methyl oxidation, why not choose a metabolite from N-deethylation?

Lines 95-96, please add approval information, such as number, date.

Line 121, please cite references for the literature mentioned here.

Please give purchase information for standards of DEET, DCBA and DHMB.

Please make a table for QA/QC results (i.e., recovery rates).

Reviewer #2: Important and innovative study. Very good scientific contribution. The authors demonstrate knowledge about the subject. Well explored article. All figures should be improved. The presentation of tables could be improved. It would be important to add their hypothesis in the introduction.

Reviewer #3: Some suggestions/additional comments to the authors:

There are some abbreviations throughout the manuscript. Consider making a list of abbreviations and meanings.

What is the toxic relevance of DEET and its metabolites, DCBA and DHMB ?

Line 172 - Analytical methods for urine samples – was based on which reference ?

Was any internal standard used in the analysis?

Pilot study

Line 299 – of the 14 children ... (age?)

8 participants only? Were there not 13 remaining participants? It didn't seem clear to me...

Reviewer #4: The manuscript intitled "Development of an observational exposure human biomonitoring study to assess

Canadian children's DEET exposure during protective use" by Gibson et. presents an interesting issue, however needs profound modifications, including statistical and sample data, as foolow below:

1) Why the authors chose the children age. Besides, sometimes the authors refers as 6-12 years old, and sometimes 7-12 years old;

2) Phisiologically, these chosen ages may reflect different toxicological effects. These issue is fundamental and underlies the importance of the study;

3) The authors should stratify the samples (i.e., 7-10 and 11-13), if the commentary above is positive physiologically;

4) Line 55-56: such kind of justification should not be raised;

5) The concentration of repelent applied should be considered in the measurement of urine DEET and metabolites concentration, since that higher concentrations induces higher concentrations in urine (first-order knetics processes);

6) What is the novelty of the work? This issue was not clear;

7) The title did not reflects the study;

8) The number of the license of Health Canada’s Research Ethics Board approvation;

9) What means QA/QC samples?

10) What the consequences of body system bioavaliablity of DDET and its metabolites for Human health?

11) What the secure blood levels of DEET and its metabolites? Are the metabolites bioactive compounds toxic for the Human?

12) Conclusion needs to be more direct, with the conclusion and importance of the study.

6. PLOS authors have the option to publish the peer review history of their article (what does this mean?). If published, this will include your full peer review and any attached files.

Reviewer #1: No

Reviewer #2: No

Reviewer #3: No

Reviewer #4: No

---

## [Author Response · Author response to Decision Letter 0]

14 Mar 2022

Dear Editor,

Please find our revised manuscript and additional documentation in response to the peer review comments attached to this response letter in the PLOS ONE Editorial System.

1. Adjusted file names to meet PLOS ONE requirements and ensured all documents follow PLOS ONE formatting guidelines.

2. Data Availability Statement: 

We have engaged our Privacy Management Division at Health Canada on the question of data availability. After reviewing the data that is associated with this publication, they have concluded that the data could be considered personal information, so Canada’s Privacy Act applies. This means that the data cannot be released to the Journal since this disclosure is not for the purposes for which it was collected nor is it a consistent use nor is it disclosed with consent, as required by the Privacy Act.

As such, the minimal data set that I have provided in association with the revised manuscript contains only the results from the quality assurance/quality control (QA/QC) testing that occurred during the study, and which is discussed in the article. Please see file Supplementary Information QAQC data.xlsx for these data.

3. The full ethics statement was moved to be the first paragraph of the Methods section.

Please find below a table of responses to reviewers comments.

Sincerely,

Jennifer Gibson

Review Comments to the Author

Reviewer #1

Overall comment

[…] the objective of this study […] is not very clear

The objective of the study is more clearly stated.

Overall comment

The authors established three exposure scenarios for data analysis, but did not conduct a good comparison between them and suggest better ones (at least not seen in Abstract, but in Conclusion). The authors need to indicate the chosen scenarios explicitly by showing some good approaches of comparison, and make this paper more valuable.

The purpose of this paper is to report how the study was executed and validate the study approach. Due to the volume of data generated and the number of relationships we needed to test, there will be a follow up paper that presents more in depth analyses of the results, including statistically comparing the urine concentrations over time in various lag scenarios.

The rationale for selecting 0, 2, and 4 hour time lags between insect repellent application and urine sampling times is a biological one, more than a statistical one. The literature suggests that a time lag exists between absorption and excretion [6, 25, 26]. This is elaborated upon in the Methods section under the Analytical decisions section.

Since the insect repellent application times were not dictated, we needed to put the urine sampling in context with the insect repellent application timing. This is why we present the construction of the three time lag scenarios.

Line 37

Please give full names of DHMB and DCBA here

Added.

Overall comment

DHMB is an intermediate product on the way to DCBA, and thus inconsistent recovery rates for it are acceptable. Other than DCBA, a final product of ring methyl oxidation, why not choose a metabolite from N-deethylation?

While DHMB is an intermediate product and therefore we did not expect to recover high concentrations of it in the urine, the inconsistent and low recovery rates for DHMB in our quality assurance samples, which should have been fixed, did not give us confidence in the participant sample recoveries. 

We chose these metabolites based on previous human biomonitoring in urine undertaken by the United States National Health and Nutrition Examination Survey (NHANES), which reports DEET, DHMB and DCBA. It would be very interesting for a future study to investigate a metabolite expected from the other metabolite pathway.

Lines 95-96

Please add approval information, such as number, date.

Added.

Line 121

Please cite references for the literature mentioned here.

Added.

Line 154

Please give purchase information for standards of DEET, DCBA, and DHMB.

Added the purchase information.

Line 353

Please make a table for QA/QC results (i.e., recovery rates).

Added tables of QA/QC results to the Supplementary Material.

Reviewer #2

Overall comment

All figures should be improved.

All figures were originally submitted after using the PACE tool. It is unclear what needs to be improved.

Overall comment

The presentation of tables could be improved.

The tables are formatted to meet the PLOS ONE requirements. It is unclear what needs to be improved.

Overall comment

It would be important to add their hypothesis in the introduction.

The objective of the study was more clearly indicated.

Reviewer #3

Overall comment

There are some abbreviations throughout the manuscript. Consider making a list of abbreviations and meanings.

Created a list of abbreviations for the Supplementary Material.

Overall comment

What is the toxic relevance of DEET and its metabolites, DCBA and DHMB?

DEET is a controlled compound under Canada’s Pest Control Products Act. It has been shown that DEET can be used as a safe and effective personal insect repellent. However, as children are considered a vulnerable population, the guidelines for DEET use by children are necessarily more restrictive than those established for adults.

It is known that DEET can be absorbed through the skin when used, and the body metabolises it and excretes it as several metabolites, including DHMB and DCBA. This has not been well studied in children, therefore this study seeks to generate data to better understand the relationship between real world application and excretion in children using DEET as they would normally to protect against insect bites while at overnight summer camp.

Line 172

Analytical methods for urine samples – was based on which reference? Was any internal standard used in the analysis?

Reference added. Internal reference samples were used, along with field QA/QC samples. 

Line 299

Pilot study – of the 14 children … (age?) 8 participants only? Were there not 13 remaining participants? It didn’t seem clear to me…

Reorganized the paragraph.

One of the pilot study children didn’t participate at all and was dropped from the pilot study. 

Of the 13 children who did participate in filling in the activity journal for the pilot study, only 8 of those children provided urine samples. We were able to gather 10 urine samples from those 8 children.

Reviewer #4

Overall comment

1) Why the authors chose the children age. Besides, sometimes the authors refers as 6-12 years old, and sometimes 7-12 years old.

The age range was chosen to try to recruit younger children (6 to 12 years old) who would be old enough to be able to engage with the study material while still providing us with a younger cohort, where the data gap is most pronounced. 

We were originally attempting to recruit children between 6 and 12 years of age, but we were unable to recruit 6 year olds at all, either because the overnight summer camps didn’t have children that age at camp, or because of lack of interest from the parents or children. We successfully recruited children between the ages of 7 and 13 years of age.

The original intended age range is discussed as it was a pertinent shift in the study parameters that we had to make in order accomplish the study.

Overall comment

2) Phisiologically, these chosen ages may reflect different toxicological effects. These issue is fundamental and underlies the importance of the study; 

For other chemicals, it is known that children have different metabolic capabilities than adults and that is why they are risk assessed as vulnerable populations. This study seeks to gather data about how children normally use DEET and what excretion results from use. 

While it is likely that some of the older children were beginning puberty, we didn’t gather that data to be able to stratify accurately according to biological development. For the purposes of our study, we are assuming that the children within the age range are similar enough, physiologically.

Overall comment

3) The authors should stratify the samples (i.e., 7-10 and 11-13), if the commentary above is positive physiologically.

In addition to the response above regarding age stratification - unfortunately, we didn’t reach our recruitment target (n=200) so our sample population is already smaller than we hoped. If we stratified, we would have (n=52) in the 7-10 age group and (n=71) in the 11-13 age group. 

With the amount of existing variability in the data due to the less controlled nature of the study, we felt that it was better to keep all the age groups together. A more controlled study with a larger sample size would be a better forum to be able to see those differences.

Line 55-56

4) Line 55-56: such kind of justification should not be raised.

Removed.

Overall comment

5) The concentration of repelent applied should be considered in the measurement of urine DEET and metabolites concentration, since that higher concentrations induces higher concentrations in urine (first-order knetics processes)

The concentration of the repellent was tested for its impact on urine concentrations. This is being discussed in a follow-up paper.

Overall comment

6) What is the novelty of the work? This issue was not clear.

To our knowledge, this is the first, free living, real-world observational exposure and human biomonitoring study involving children using DEET-based insect repellents. This has been emphasized in the introduction.

Overall comment

7) The title did not reflects the study

This paper is intended to publish the approach we took to achieve a free living, real-world observational exposure and human biomonitoring study involving children using DEET-based insect repellents. This includes creating the time lag scenarios to take into account metabolism and excretion after dermal absorption in this uncontrolled study. This is why the title reflects the “development” of the project. A second paper is being prepared which will more fully investigate all the relationships we found in the results.

Overall comment

8) The number of the license of Health Canada’s Research Ethics Board approvation.

Added.

Overall comment

9) What means QA/QC samples?

Ensured the first use of the acronym has the full term preceding it.

Overall comment

10) What the consequences of body system bioavaliabity of DDET and its metabolites for Human health?

This is still under investigation by the wider scientific community. 

At this point, DEET can be used safely as an effective personal insect repellent with no apparent ill effects on human health, if guidelines are followed. But as DEET is an exogenous chemical, not essential to life, it is still risk managed by limiting exposure to the minimum effective concentration through guidelines set by Health Canada. 

Our study does not investigate the effects of DEET usage.

Overall comment

11) What the secure blood levels of DEET and its metabolites? Are the metabolites bioactive compounds toxic for the Human?

There are no blood or urine guidelines for DEET or its metabolites. 

Toxicologically, DEET is generally considered low to very low toxicity. The toxicity of the metabolites is still under investigation, but a recent paper by Segal et al (2017) was not able to find an association between urine DCBA concentrations and male infertility. This sort of investigation is beyond the scope of the current study, however.

Overall comment

12) Conclusion needs to be more direct, with the conclusion and importance of the study.

Edited.

---

## [Decision Letter · Decision Letter 1]

28 Apr 2022

Development of an observational exposure human biomonitoring study to assess Canadian children's DEET exposure during protective use.

PONE-D-21-34278R1

Dear Dr. Gibson

We’re pleased to inform you that your manuscript has been judged scientifically suitable for publication and will be formally accepted for publication once it meets all outstanding technical requirements.

Kind regards,

José Luiz Fernandes Vieira

Academic Editor

PLOS ONE

Reviewers' comments:

Reviewer's Responses to Questions

**Comments to the Author**

1. If the authors have adequately addressed your comments raised in a previous round of review and you feel that this manuscript is now acceptable for publication, you may indicate that here to bypass the “Comments to the Author” section, enter your conflict of interest statement in the “Confidential to Editor” section, and submit your "Accept" recommendation.

Reviewer #1: All comments have been addressed

Reviewer #3: All comments have been addressed

2. Is the manuscript technically sound, and do the data support the conclusions?

Reviewer #1: (No Response)

Reviewer #3: Yes

3. Has the statistical analysis been performed appropriately and rigorously? 

Reviewer #1: (No Response)

Reviewer #3: Yes

4. Have the authors made all data underlying the findings in their manuscript fully available?

Reviewer #1: (No Response)

Reviewer #3: Yes

5. Is the manuscript presented in an intelligible fashion and written in standard English?

Reviewer #1: (No Response)

Reviewer #3: Yes

6. Review Comments to the Author

Reviewer #1: (No Response)

Reviewer #3: (No Response)

7. PLOS authors have the option to publish the peer review history of their article (what does this mean?). If published, this will include your full peer review and any attached files.

Reviewer #1: No

Reviewer #3: No

---

## [Editor Report · Acceptance letter]

27 Jul 2022

PONE-D-21-34278R1 

Development of an observational exposure human biomonitoring study to assess Canadian children’s DEET exposure during protective use. 

Dear Dr. Gibson:

I'm pleased to inform you that your manuscript has been deemed suitable for publication in PLOS ONE. Congratulations! Your manuscript is now with our production department. 

Kind regards, 

on behalf of

Dr. José Luiz Fernandes Vieira 

Academic Editor

PLOS ONE